**Data Availability Statement:** All relevant data are within the paper and its Supporting Information files.

# Long-term exposure to low concentrations of polycyclic aromatic hydrocarbons and alterations in platelet indices: A longitudinal study in China

Jing Cui[1], Ting Zhang[2], Chao Zhang[2], Zhenwei Xue[2], Durong Chen[1], Xiaona Kong[2], Caili Zhao[1], Yufeng Guo[2], Zimeng Li[1], Xiaoming Liu[2], Jiefang Duan[1], Wenjie Peng[1], Xiaolin Zhou[2]*, Hongmei Yu[1]*

1 Department of Health Statistics, Shanxi Provincial Key Laboratory of Major Diseases Risk Assessment, School of Public Health, Shanxi Medical University, Taiyuan, People's Republic of China, 2 Department of Radiological and Environmental Medicine, State Environmental Protection Key Laboratory of Environment and Health (Taiyuan), China Institute for Radiation Protection (CIRP), Taiyuan, Shanxi, China

* yu@sxmu.edu.cn (HY); xiaolin0824@sohu.com (XZ)

## Abstract

Long-term exposure to low polycyclic aromatic hydrocarbon (PAH) concentration may ave detrimental effects, including changing platelet indices. Effects of chronic exposure to low PAH concentrations have been evaluated in cross-sectional, but not in longitudinal studies, to date. We aimed to assess the effects of long-term exposure to the low-concentration PAHs on alterations in platelet indices in the Chinese population. During 2014–2017, we enrolled 222 participants who had lived in a village in northern China, 1–2 km downwind from a coal plant, for more than 25 years, but who were not employed by the plant or related businesses. During three follow-ups, annually in June, demographic information and urine and blood samples were collected. Eight PAHs were tested: namely 2-hydroxynaphthalene, 1-hydroxynaphthalene, 2-hydroxyfluorene, 9-hydroxyfluorene (9-OHFlu), 2-hydroxyphe-nanthrene (2-OHPh), 1-hydroxyphenanthrene (1-OHPh), 1-hydroxypyrene (1-OHP), and 3-hydroxybenzo [a] pyrene. Five platelet indices were measured: platelet count (PLT), platelet distribution width (PDW), mean platelet volume (MPV), platelet crit, and the platelet-large cell ratio. Generalized mixed and generalized linear mixed models were used to estimate correlations between eight urinary PAH metabolites and platelet indices. Model 1 assessed whether these correlations varied over time. Models 2 and 3 adjusted for additional personal information and personal habits. We found the following significant correlations: 2-OHPh (Model1 $\beta_1$ = 18.06, Model2 $\beta_2$ = 18.54, Model $\beta_3$ = 18.54), 1-OHPh ($\beta_1$ = 16.43, $\beta_2$ = 17.42, $\beta_3$ = 17.42), 1-OHP($\beta_1$ = 13.93, $\beta_2$ = 14.03, $\beta_3$ = 14.03) with PLT, as well as 9-OHFlu with PDW and MPV (odds ratio or Model3 $OR_{PDW}$[95%CI] = 1.64[1.3–2.06], $OR_{MPV}$[95%CI] = 1.33[1.19–1.48]). Long-term exposure to low concentrations of PAHs, indicated by2-OHPh, 1-OHPh, 1-OHP, and 9-OHFlu, as urinary biomarkers, affects PLT, PDW, and MPV. 9-OHFlu increased both PDW and MPV after elimination of the effects of other PAH exposure modes.

**Funding:** The work has been approved by the Institutional Review Board of China Institute for Radiation Protection (CIRP). Ethics approval number: 21111011101EHSM(2019)SX-03. Open fund of National Key Laboratory for environmental protection, radiation environment and health. Ethics approval number:YP21020203.

**Competing interests:** The authors have declared that no competing interests exist.

## Introduction

Polycyclic aromatic hydrocarbons (PAHs) are toxic organic pollutants with at least two fused aromatic rings of varying origin. Outdoors are generated by the incomplete combustion of coal, petroleum, wood, tobacco, and organic macromolecular compounds [1]. Indoors, PAHs are typically derived from smoking and carbon-baked foods [2]. The acute effects of PAHs on human health depend on the concentration, duration, and route of exposure. Multiple studies have focused on occupational and high-concentration exposures, which affect the nasal tissues, red blood cells, platelets, white blood cells, uterus, hair follicles, brain, spleen, placenta, liver, lungs, and kidneys [3]. Chronic effects include immunotoxicity, cytotoxicity, immune dysfunction, dyslipidemia [4], asthma [5], fetal dysplasia [6], and cardiovascular disease [7]. For example, high-dose exposure to vehicular combustion products has been reported to increase platelet counts [8] and exert other effects on platelet activity [9]. In rabbits, high-dose exposure to PAHs affects the synthesis of thromboxane B2 [10].

Long-term exposure to low concentrations of PAHs may cause low-grade inflammation [11] due to alterations in the platelet index [12]. The effects of chronic exposure to low concentrations of PAHs have also been estimated from cross-sectional research for both adults and children [13]; however, these correlations have not been proven in longitudinal analysis, which has the advantage of allowing estimation of the effect of long-term exposure.

Risk assessment of environmental pollution relies on dose–response relationships [3], and biomarkers have been suggested to be reliable epidemiological tools. The breakdown products of PAHs are excreted chiefly in urine [1, 6, 7, 14].

Thus, we assessed the effects of long-term exposure to low-concentration PAHs and alterations in platelet indices in the Chinese population. Longitudinal datasets were collected to study the effects of long-term exposure to low concentrations of PAHs on platelet indices among adults who were not occupationally exposed. We then estimated the correlations among eight representative urinary PAH metabolites (UPAHMs) and five platelet indices, under the effect of covariates, aiming to assess low-concentration exposure to PAHs and alterations in platelet indices in the Chinese population based on a longitudinal dataset using UPAHM as biomarkers of exposure.

## Materials and methods

### Study population

We enrolled 418 participants from northern China [15] between 2014 and 2017. All participants had resided in a village located 1–2 km downwind from a coal plant for more than 25 years, but were not employed by the plant or related businesses. As the residences were close to each other in the village, participant exposure to automobile exhaust was not considered. During three annual waves of follow-up of the 2014, 2015, 2016, and 2017 groups, in June (non-heating season) each year, demographic information was collected via questionnaires through in-person interviews conducted by rigorously trained interviewers. Urine and blood samples were collected on the same morning at each of the three follow-up time-points. Complete records were available for 222 subjects (Fig 1).

Participants were grouped by their year of enrolment: 59 participants in 2014, as the first group, 71 in 2015, as the second group, 53 in 2016, as the third group, and 39 in 2017, as the fourth group. All subjects provided informed consent for participation and for storage and use of their blood and urine samples.

### Measurement of urinary metabolites

Morning urine samples were collected from each participant in clean polypropylene tubes in June during the three consecutive annual follow-ups. All urine samples were stored at −20˚C

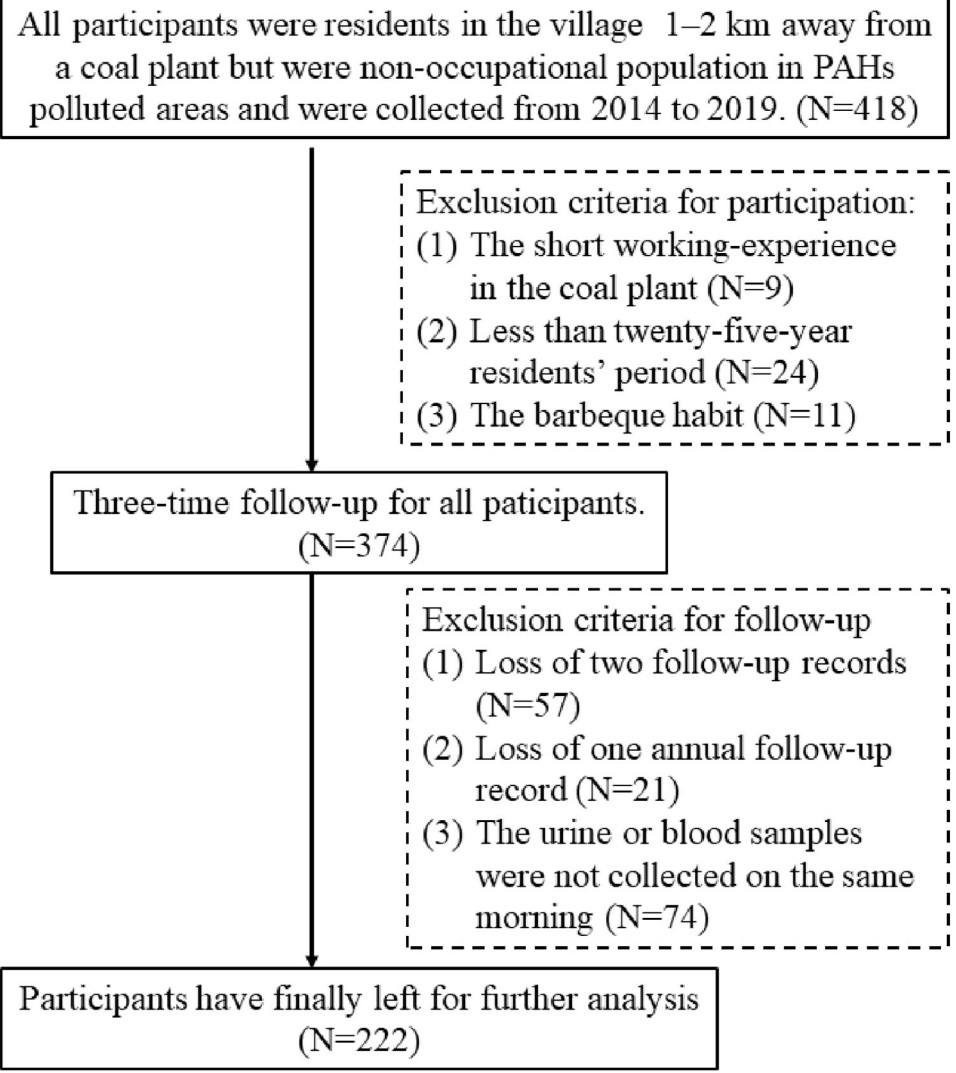

**Fig 1. The include and exclude processes of the participants.**

until used. The limits of detection of PAHs ranged from 0.1 to 0.9 µg/L and default values were replaced by 50% of the limit of detection. We tested samples for eight PAHs: 2-hydroxynaphthalene (2-OHNa), 1-hydroxynaphthalene (1-OHNa), 2-hydroxyfluorene (2-OHFlu), 9-hydroxyfluorene (9-OHFlu), 2-hydroxyphenanthrene (2-OHPh), 1-hydroxyphenanthrene (1-OHPh), 1-hydroxypyrene (1-OHP), and 3-hydroxybenzo[a]pyrene (3-OHBaP). The testing process has been previously described [16]. Briefly, urine samples were hydrolyzed with β-glucuronidase/sulfatase (Roche, Basel, Switzerland) and purified using C18 cartridges (surface area: 525 m$^2$/g per cartridge; average particle size: 52.1 µm, Supelco, Inc., Bellefonte, PA, USA). Next, a 400-µL extract was produced by condensation with a dry N2 purge. High-performance liquid chromatography (Waters-2695, Waters Ltd., Milford, MA, USA) with a fluorescence detector. The linearity (expressed as the R-value), mean relative standard deviation, and mean recovery rate of the samples were 0.999%–1.0000%, 0.70%–8.36%, and 81.83%–123.75%, respectively. To avoid fluctuation of substances in the urine with the amount of urine excreted, the UPAHMs were calibrated using urinary creatinine.

## Measurement of platelet indices

Fasting blood samples (15 mL) were collected from each participant using standard methods [17] and were analyzed for leukocytes, erythrocytes, thrombocytes, and immunoglobulin indices using a Hemaray 86 automatic hematological analyzer (Rayto Co., Shen Zhen, China). Five platelet indices were measured: platelet count (PLT), platelet distribution width (PDW), mean platelet volume (MPV), platelet crit, and platelet-large cell ratio (P-LCR).

The measurement of urinary metabolites and platelet indices was also applied in a previous study [18].

## Covariates

Age, sex, weight, and height were used as covariates. Body mass index (BMI) was calculated as weight divided by height. The participants were grouped by age ($\leq$ 60 and > 60 years). Habitual smokers were defined as those who smoked at least one cigarette per day for at least 6 months [19]. Second-hand smoke exposure was defined as the presence of an adult who smoked more than one pack per week. Alcohol consumption was defined as a history of alcohol consumption for more than 1 year. Considering that high-frequency intake of barbecued foods increases PAH exposure [20] and that barbecue was not the primary cooking method in this village, the 11 participants who reported barbecued foods were excluded from the analysis.

## Statistical analyses

First, we used univariate statistical analysis to evaluate demographic characteristics and UPAHM levels, and their changes between sampling points. We used the Shapiro–Wilk test to assess normality, and analysis of variance, least significant difference, chi-squared test, Wilcoxon's rank-sum test, and the Kruskal–Wallis H test to identify differences between the groups. Next, we used Spearman's correlation analysis and K-means cluster methods to determine whether demographic information correlated with mean changes in the UPAHM. To identify changes in thrombocyte indices and UPAHM over time, we used multivariate analysis of variance for repeated measurements [21] and traditional analysis of variance for variables that met and did not meet sphericity assumptions, respectively. We then used general mixed models [22] to evaluate the correlations between the UPAHM levels and thrombocyte indices. After excluding indices without statistical significance, we constructed generalized linear mixed models [23] to determine whether the correlations had a linear distribution. To estimate the effect of PAHs on the thrombocyte indices, we constructed three adjusted models. Model 1 was a time model with "group" and "ID" as the covariates. Model 2 was adjusted for "group," "ID," "age," "sex," and "body mass index." Model 3 included all the variables adjusted for in Model 2, in addition to smoking status, exposure to secondhand smoke, and alcohol consumption. We also estimated the Akaike Information Criterion (AIC) [24] and Bayesian Information Criterion (BIC) [25] to assess the models. The AIC transforms the penalized likelihood into a negative log-likelihood plus a penalty term in Eq 1, where k stands for the number of free parameters. The BIC exerts a higher penalty than the AIC for model overfitting (Eq 2). As long as the true model is a candidate model, the model with the minimum BIC exhibits the best performance. If the true model is not a candidate model, the minimum AIC exhibits the best performance [26]. The roadmap of the model-setting process is shown in Fig 2.

$$\text{AIC} = -2\text{logL}(\hat{\theta}) + 2\text{k} \tag{1}$$

$$\text{BIC} = -2\text{logL}(\hat{\theta}) + \ln(\text{k}) \tag{2}$$

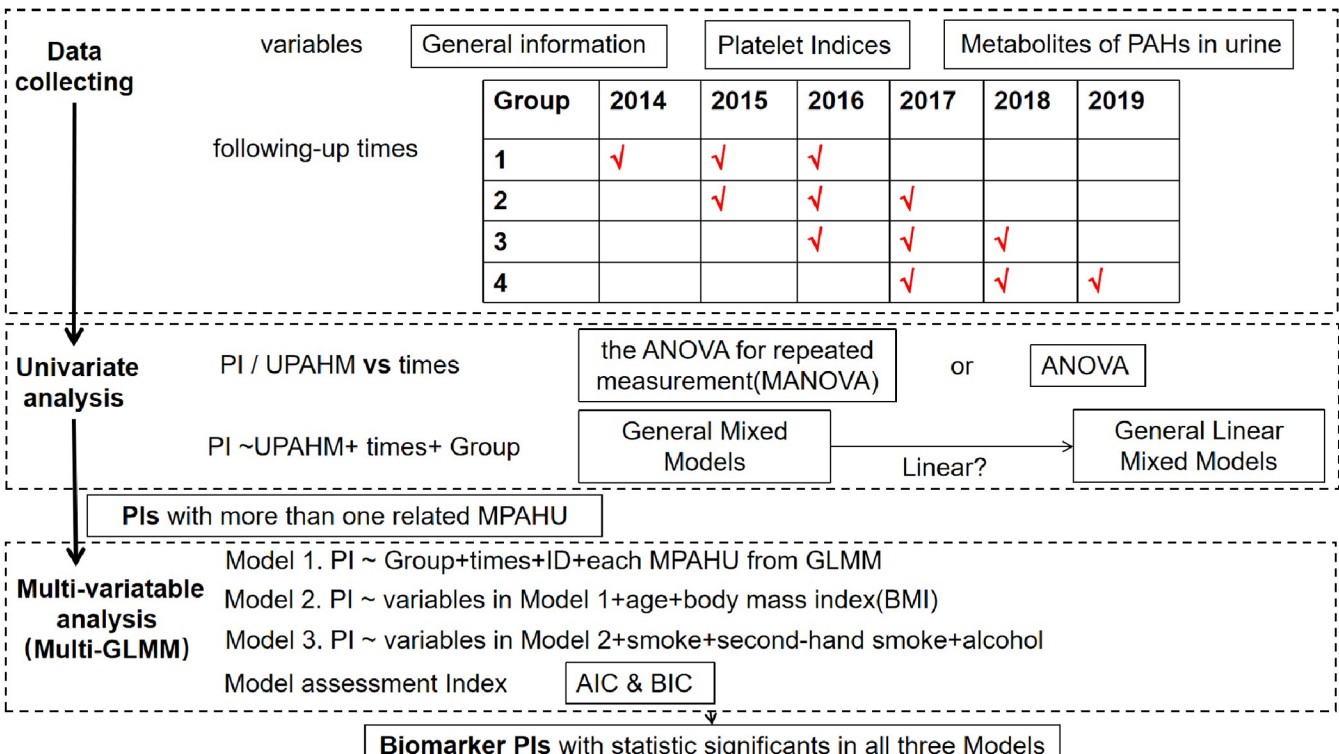

**Fig 2. The statistical analysis process in the correlation discovery.**

All data were double-entered into EpiData 3.0, with a consistency of 99.9%, and all the models were fitted using SAS 9.4 (SAS Institute, Cary, NC, USA). The first section of the Supplementary Materials provides more details on the model fitting.

## Ethical approval

This research was supported by the China Institute for Radiation Protection under license. Informed consent was obtained from all participants and/or their legal guardians. Research involving human participants was performed in accordance with the tenets of the Declaration of Helsinki. The license and ethical approval have been uploaded as related files.

## Scientific application of the methods

As there is currently no gold standard for the estimation of UPAHM extraction, we extracted them following the patent named "The established method to analyze eight OH-PAHs in urine simultaneously" (202010201466.3), the details of which were uploaded as the relevant document into the submission system. When R > 0.999, the patent is still under application. The blood indices followed standard methods [17].

## Results

### Characteristics of study participants

Participants' ages ranged from 55 to 65 years, and the age range did not vary by year of recruitment into the study. More than half of the participants were females. More than half of the participants (56%) were overweight, with no specific difference between the groups. Changes in

**Table 1. Basic information of 222 participants.**

| Group | First | Second | Third | Fourth | Statistic | P-value |
|---|---|---|---|---|---|---|
| No. of participants | 59 | 71 | 53 | 39 | | |
| Age(years) | 61.50 (57.25, 65.00) | 61.00 (55.00, 64.00) | 63.00 (60.00, 66.00) | 62.00 (57.25, 65.00) | 4.595[†] | 0.204 |
| Gender (Male/Female) | 23/37 | 24/47 | 13/39 | 12/27 | 2.374[*] | 0.498 |
| Height (cm) | 162.00 (155.00, 167.00) | 164.00 (158,00, 168.75) | 160.00 (156.00, 166.00) | 160.00 (158.00, 168.00) | 1.915[†] | 0.590 |
| Weight (kg) | 65.00 (58.50, 71.75) | 65.00 (57.13, 70.00) | 62.00 (56.00, 68.00) | 65.00 (58.05, 74.50) | 1.643[†] | 0.650 |
| BMI(kg/m$^2$) | 24.86±2.85 | 24.35±3.50 | 24.13±4.11 | 25.70±4.39 | 3.036[§] | 0.386 |
| $\Delta_{12}$2-OHNa(ng/g·cr$^{-1}$) | 0.63 (0.26, 1.16) | 0.66 (0.24, 1.09) | 0.58 (0.25, 1.13) | 0.48 (0.24, 1.08) | 1.013[†] | 0.798 |
| $\Delta_{23}$2-OHNa(ng/g·cr$^{-1}$) | 0.66 (0.27, 1.16) | 0.55 (0.20, 1.08) | 0.45 (0.25, 0.95) | 0.60 (0.14, 1.09) | 1.757[†] | 0.624 |
| $\Delta_{12}$1-OHNa(ng/g·cr$^{-1}$) | 0.80 (0.19, 1.69) | 0.70 (0.30, 1.48) | 0.54 (0.26, 1.71) | 1.11 (0.41, 1.81) | 1.140[†] | 0.767 |
| $\Delta_{23}$1-OHNa(ng/g·cr$^{-1}$) | 0.55 (0.23, 1.53) | 0.67 (0.25, 1.67) | 1.10 (0.35, 1.89) | 1.31 (0.33, 1.98) | 3.508[†] | 0.320 |
| $\Delta_{12}$2-OHFlu(ng/g·cr$^{-1}$) | 0.45 (0.16, 0.96) | 0.45 (0.20, 0.94) | 0.27 (0.03, 0.70) | 0.31 (0.08, 0.62) | 6.025[†] | 0.110 |
| $\Delta_{23}$2-OHFlu(ng/g·cr$^{-1}$) | 0.43 (0.18, 0.92) | 0.35 (0.11, 0.76) | 0.41 (0.07, 0.78) | 0.22 (0.04, 0.83) | 2.382[†] | 0.497 |
| $\Delta_{12}$9-OHFlu(ng/g·cr$^{-1}$) | 0.55 (0.01, 1.31) | 0.72 (0.14, 1.36) | 0.43 (0.00, 0.99) | 0.58 (0.00, 1.20) | 3.560[†] | 0.313 |
| $\Delta_{23}$9-OHFlu(ng/g·cr$^{-1}$) | 0.65 (0.15, 1.38) | 0.62 (0.15, 1.29) | 0.67 (0.00, 1.28) | 0.77 (0.01, 1.20) | 0.706[†] | 0.872 |
| $\Delta_{12}$2-OHPh(ng/g·cr$^{-1}$) | 0.55 (0.27, 1.13) | 0.56 (0.22, 0.83) | 0.53 (0.28, 0.97) | 0.39 (0.17, 0.85) | 3.074[†] | 0.380 |
| $\Delta_{23}$2-OHPh(ng/g·cr$^{-1}$) | 0.53 (0.22, 0.95) | 0.62 (0.31, 1.00) | 0.50 (0.13, 0.87) | 0.63 (0.21, 1.10) | 1.011[†] | 0.799 |
| $\Delta_{12}$1-OHPh(ng/g·cr$^{-1}$) | 0.43 (0.19, 0.99) | 0.57 (0.23, 0.82) | 0.44 (0.13, 0.92) | 0.48 (0.15, 0.89) | 0.592[†] | 0.898 |
| $\Delta_{23}$1-OHPh(ng/g·cr$^{-1}$) | 0.56 (0.15, 0.82) | 0.40 (0.17, 0.93) | 0.52 (0.16, 0.94) | 0.48 (0.14, 0.95) | 0.826[†] | 0.843 |
| $\Delta_{12}$1-OHP(ng/g·cr$^{-1}$) | 0.44 (0.08, 1.06) | 0.47 (0.16, 0.87) | 0.39 (0.10, 0.86) | 0.20 (0.09, 0.46) | 4.009[†] | 0.261 |
| $\Delta_{23}$1-OHP(ng/g·cr$^{-1}$) | 0.40 (0.14, 0.87) | 0.50 (0.14, 1.03) | 0.25 (0.11, 0.63) | 0.33 (0.14, 0.64) | 3.888[†] | 0.274 |
| $\Delta_{12}$3-OHBaP(ng/g·cr$^{-1}$) | 0.75 (0.00, 1.35) | 0.68 (0.00, 1.27) | 0.61 (0.08, 1.18) | 0.72 (0.02, 1.48) | 1.112[†] | 0.774 |
| $\Delta_{23}$3-OHBaP(ng/g·cr$^{-1}$) | 0.69 (0.00, 1.53) | 0.69 (0.00, 1.18) | 0.50 (0.00, 1.03) | 0.76 (0.31, 1.40) | 3.457[†] | 0.326 |

Note, $\alpha$ = 0.05, [†] following the statistic values stand for the data in the four groups were not all satisfied the normality, and the differences among the groups were analyzed by the Kruskal-Wallis H tests with median, first, and third quantiles for statistic description. [§] following the statistic values stand for the data in the four groups satisfied the normality, and the ANOVA analyzed the differences among the groups with mean and standard deviation for statistic description. [*] stands for the Chi-square test were applied for the analysis with frequency.

urinary metabolite levels between sampling time-points did not differ by the year of recruitment (Table 1).

## Demographic information and urinary metabolite levels

Spearman correlation coefficients for the relationship between urinary PAH metabolite levels and platelet indices differed from the null hypothesis (P < 0.05) but were smaller than 0.1. The highest coefficient was 0.15, calculated for the relationship between the levels of 2-OHFlu and platelet distribution width. Second-hand smoke exposure and age were clustered with three UPAHMs (1-hydroxynaphthalene, 9-OHFlu, and 33-hydroxybenzo[a]pyrene; S1 Fig). Although their correlations lacked statistical significance, we included age and second hand smoke exposure as covariates for further model fitting.

## Variations in urinary metabolite levels over time

We analysed variations over time in UPAHM levels and platelet indices. To eliminate the effect of recruitment year, we first grouped participant data by year of recruitment into the study. All participants were healthy, and we did not find any changes in platelet indices between consecutive years. The UPAHM levels were log-transformed before testing; therefore, some of the means were negative. The levels of five UPAHMs measured over three years varied among the

participant groups recruited in 2014 and those recruited in 2015. The levels of 2-OHNa and 1-OHPh varied over time in groups recruited in the first and second groups. The levels of 2-OHPh varied in the first group, and those of 2-OHFlu and 1-OHP varied in the second group. S5 Table shows the variation in platelet indices and UPAHMs.

### Urinary metabolite levels and platelet indices

To assess the effect of PAH exposure on platelet indices, we fitted general mixed models to the data to estimate fixed effects. Eight correlations were statistically significant: correlations of 1-OHNa, 2-OHFlu, 2-OHPh, 1-OHPh, and 1-OHP with PLT, and correlations of 9-OHFlu with PDW, MPV, and P-LCR (Table 2). S1 Table lists the random effects of the general mixed models.

### Dose-response associations in exposure and platelet

We fitted three general linear mixed models (GLMMs) using eight correlations for the dose–response analysis using various covariates. We used GLMM with a continuous data as well as with UAPHM levels categorized into four groups, and estimated how these exposures would affect platelet indices. We found a linear relationship of the categorized levels of 1-OHNa, 2-OHPh, 1-OHPh, 1-OHP, with PLT, whereas the continuous data of 2-OHFlu, 2-OHPh, 1-OHPh, and 1-OHP were significantly correlated (Table 3). We identified UPAHMs showing significance as biomarkers in both analyses of continuous and level category analyses: 2-OHPh, 1-OHPh, and 1-OHP satisfied this requirement. According to the AIC and BIC, Model 3, which contained the most covariates, performed the best. S2 Table lists the odds ratios and 95% confidence intervals (CI) for the covariates.

Urinary 9-OHFlu levels were correlated with PDW, MPV, and P-LCR in both continuous forms and as level categories. Analyzing continuous data revealed that 9-OHFlu (coefficients for Model 1: $\beta_1$[P value] = 0.48 [< 0.001], Model 2 $\beta_2$[P value] = 0.49 [< 0.001], Model 3 $\beta_3$[P value] = 0.49 [< 0.001]) correlated with PDW and with MPV ($\beta_1$[P value] = 0.28 [< 0.001], $\beta_2$[P value] = 0.28 [< 0.001], $\beta_3$[P value] = 0.28 [< 0.001]), and with P-LCR ($\beta_1$[P value] = 0.58 [0.458], $\beta_2$[P value] = 0.57 [0.463], $\beta_3$[P value] = 0.57 [0.463]) in all three models, all of which had an increasing effect on platelet indices.

GLMM showed that secondhand smoke significantly impacted the association of 9-OHFlu with platelet indices. Subgroup analysis by exposure to secondhand smoke (yes/no) was

**Table 2. The fixed effect among platelet indices and UPAHM based on GMMs.**

| UPAHM | PLT/F1(%) | PDW(fL) | MPV(fL) | PCT(%) | P-LCR(%) |
|---|---|---|---|---|---|
| 2-OHNa(ng/g·cr$^{-1}$) | 0.34 (0.5627) | 3.03 (0.0826) | 3.07 (0.0806) | 2.36 (0.1253) | 0.21 (0.6497) |
| 1-OHNa(ng/g·cr$^{-1}$) | 5.22 (0.0228)* | 0.23 (0.6332) | 0.13 (0.7201) | 0.44 (0.5060) | 0.10 (0.7527) |
| 9-OHFlu(ng/g·cr$^{-1}$) | 3.75 (0.0534) | 6.01 (0.0146)* | 6.26 (0.0127)* | 0.01 (0.9270) | 4.37 (0.0372)* |
| 2-OHFlu(ng/g·cr$^{-1}$) | 5.45 (0.0200)* | 0.21 (0.6469) | 0.42 (0.5165) | 0.53 (0.4652) | 1.20 (0.2744) |
| 2-OHPh(ng/g·cr$^{-1}$) | 4.71 (0.0306)* | 0.16 (0.6922) | 0.22 (0.6399) | 1.74 (0.1884) | 2.10 (0.1480) |
| 1-OHPh(ng/g·cr$^{-1}$) | 5.45 (0.0200)* | 0.31 (0.5761) | 0.41 (0.5227) | 0.77 (0.3806) | 0.62 (0.4314) |
| 1-OHP(ng/g·cr$^{-1}$) | 6.17 (0.0134)* | 0.30 (0.5862) | 0.16 (0.6854) | 0.67 (0.4133) | 2.65 (0.1040) |
| 3-OHBaP(ng/g·cr$^{-1}$) | 0.01 (0.9050) | 1.57 (0.2112) | 0.77 (0.3811) | 2.03 (0.1549) | 0.61 (0.4339) |

Note, we have settled the GMMs. Each model took each UPAHM as a fixed effect, while rectified six random effects, including UPAHM with the group and following-up time, UPAHM with following-up time, UPAHM with group, group with following-up time, group, and following-up time. The F values and P values of each UPAHM's fixed effect have been displayed in Table above, as $\alpha$ = 0.05

* stands for the fixed effect of UPAHM with the platelet indices have statistical significance.

**Table 3. Dose-reposed effect of PAH exposure on PLT.**

| Level of UPAHM | Model 1 | Model 2 | Model 3 |
|---|---|---|---|
| 1-OHNa(ng/g·cr$^{-1}$) | | | |
| Con(β[P value]) | 6.65[0.052] | 6.54[0.057] | 6.54[0.057] |
| 1 | 0.95(0.916–0.986)* | 0.951(0.916–0.987)* | 0.95(0.916–0.986)* |
| 2 | 0.945(0.9–0.993)* | 0.946(0.9–0.993)* | 0.943(0.897–0.991)* |
| 3 | 0.962(0.924–1.001) | 0.962(0.924–1.002) | 0.961(0.923–1.001) |
| 4 | 1 | 1 | 1 |
| AIC | -286.9 | -268.9 | -251 |
| BIC | -266.5 | -248.5 | -230.6 |
| 2-OHFlu(ng/g·cr$^{-1}$) | | | |
| Con(β[P value]) | 22.37[<0.001*] | 22.8[<0.001*] | 22.8[<0.001*] |
| 1 | 0.966(0.928–1.005) | 0.966(0.928–1.006) | 0.967(0.928–1.007) |
| 2 | 0.961(0.923–1) | 0.961(0.922–1) | 0.961(0.923–1.001) |
| 3 | 1.008(0.969–1.049) | 1.007(0.967–1.049) | 1.007(0.967–1.049) |
| 4 | 1 | 1 | 1 |
| AIC | -286.2 | -268.2 | -249.6 |
| BIC | -265.8 | -247.8 | -229.2 |
| 2-OHPh(ng/g·cr$^{-1}$) | | | |
| Con(β[P value]) | 18.06[<0.001*] | 18.54[<0.001*] | 18.54[<0.001*] |
| 1 | 0.941(0.904–0.979)* | 0.941(0.904–0.98)* | 0.941(0.904–0.98)* |
| 2 | 0.939(0.904–0.976)* | 0.939(0.904–0.977)* | 0.94(0.904–0.977)* |
| 3 | 0.951(0.912–0.991)* | 0.95(0.911–0.99)* | 0.949(0.91–0.99)* |
| 4 | 1 | 1 | 1 |
| AIC | -290.1 | -272.2 | -253.8 |
| BIC | -269.7 | -251.8 | -233.4 |
| 1-OHPh(ng/g·cr$^{-1}$) | | | |
| Con(β[P value]) | 16.43[<0.001*] | 17.42[<0.001*] | 17.42[<0.001*] |
| 1 | 0.958(0.92–0.997)* | 0.958(0.92–0.997)* | 0.957(0.919–0.996)* |
| 2 | 0.958(0.924–0.993)* | 0.958(0.924–0.994)* | 0.958(0.924–0.994)* |
| 3 | 0.992(0.944–1.042) | 0.991(0.943–1.041) | 0.99(0.942–1.04) |
| 4 | 1 | 1 | 1 |
| AIC | -285.4 | -267.5 | -249.2 |
| BIC | -265 | -247.1 | -228.8 |
| 1-OHP(ng/g·cr$^{-1}$) | | | |
| Con(β[P value]) | 13.93[0.001*] | 14.03[0.001*] | 14.03[0.001*] |
| 1 | 0.948(0.902–0.996)* | 0.949(0.903–0.997)* | 0.949(0.903–0.997)* |
| 2 | 0.99(0.942–1.04) | 0.992(0.944–1.043) | 0.992(0.944–1.043) |
| 3 | 0.988(0.942–1.035) | 0.989(0.944–1.037) | 0.99(0.944–1.038) |
| 4 | 1 | 1 | 1 |
| AIC | -285.4 | -267.8 | -249.4 |
| BIC | -265 | -247.3 | -228.9 |

Notes, by adding different covariants, three general linear models were settled for the dose-respond effect of the PAH exposure and the PLT. The "Con" stands for the continuous form of each UPAHM that has been included in the GLMM with the coefficient and p-value(β[P value]), while the rest "1", "2", "3", "4" were the results of the categorized UPAHM. For the three GLMM, the group and time were adjusted in Model 1, while Model 2 added extra three demographics information based on Model 1, including age, gender, and BMI. Besides, Model 3 added more habitual variables based on Model 2, including smoke, drink, and passive smoke. The Odd ratio (OR) and their 95% confidence intervals were listed in the above Table.

* stands for the UPAHM level were different from the reference level with statistical significance and $\alpha$ = 0.05. AIC and BIC for each model were listed after the levels.

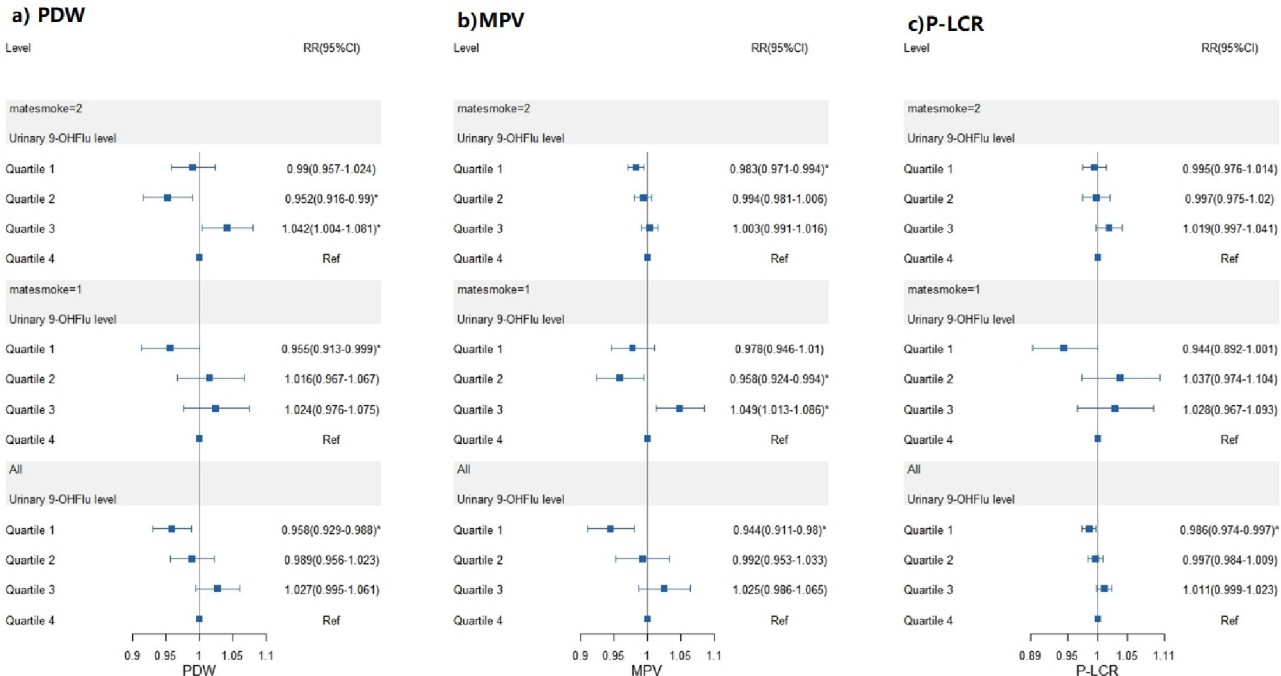

**Fig 3. The effect of the second-hand smoke to the correlations observed between 9-OHFlu levels and platelet indices.** Whether the correlations were true or confounded by exposure to second-hand smoke. "Matesmoke = 1" stands for without the effect of the second-hand smoke, while the "Matesmoke = 2" stands for the subgroup with the effect of the second-hand smoke.

conducted using Model 1 to determine whether the correlations observed between 9-OHFlu levels and platelet indices were true, or had been confounded by exposure to secondhand smoke (Fig 2). For PDW, participants without exposure to secondhand smoke exhibited a dose–response trend in the entire cohort, in which the first tested exposure level differed from the highest exposure effect (odds ratio without exposure to secondhand smoke [$OR_{without}$]: 0.955; 95%CI:0.913–0.999). For MPV, participants without exposure to second-hand smoke exhibited a dose–response trend at the second and third levels, in which the second exposure level ($OR_{without}$ = 0.958; 95%CI [0.924–0.994]) and the second exposure level ($OR_{without}$ = 1.049; 95%CI[1.013–1.086]) differed from the highest exposure level. For the platelet-large cell ratio, the subgroups showed no significant difference (Fig 3). S3 Table lists the details of the analysis and the general linear mixed models.

## Discussion

We used a longitudinal dataset to investigate the effects of exposure to low concentrations of PAHs on platelet indices, using UPAHMs as biomarkers of exposure. Five metabolites were associated with PLT: 2-OHFlu, 1-OHNa, 2-OHPh, 1-OHPh, and 1-OHP, of which all except 2-OHFlu exhibited dose–response relationships with linear correlations. Additionally, urinary levels of 9-OHFlu were linearly correlated with PDW, MPV, and P-LCR, while exposure to secondhand smoke was also a significant factor in the GLMM. An additional subgroup analysis showed that secondhand smoke exposure might bias the correlations between 9-OHFlu levels and platelet indices: 9-OHFlu had an increasing effect on PDW at the first level and MPV increased significantly with 9-OHFlu.

Platelets contribute to hemostasis and coagulation, and have been proposed as blood biomarkers in studies of inflammation and immune responses. The MPV and PDW have been

associated with coronary artery diseases, as they may reflect the size and activity of platelets in thrombosis and inflammation. The induction of thromboxane A2 aggregates platelets to sites of inflammation. Changes in the MPV have been associated with hypertrophic cardiomyopathy, pulmonary hypertension, restenosis following coronary angioplasty, acute myocardial infarction, Wiskott–Aldrich syndrome, and giant platelet disorders. PDW has been suggested to be an essential clinical factor for pulmonary arterial hypertension and dementia. Identifying changes in these platelet indices and understanding their pathogenesis are therefore clinically relevant.

Inhalation, dust, and dietary ingestion are three pathways of human exposure to PAH. Intake through diet and inhalation exceeds that via dust ingestion [27], with higher concentrations found in water than in dust [28]. Epidemiological and animal experiments have shown that PAH exposure can induce inflammatory responses and alter platelet indices. PAH monohydroxy intermediates are generated via redox-active cycling with cytochrome P450 enzymes and quinone oxidoreductase. They can affect cells and tissues and react with DNA and proteins to trigger the generation of reactive oxygen species and expression of pro-inflammatory genes. By 24 or 48 h after exposure to PAHs, increases in the pro-inflammatory cytokines interleukins-1β, -8, -10, and -12 have been reported in THP-1 macrophage-like cells, and positive associations between UPAHM levels and markers of inflammation have been noted in humans [29, 30]. In this study, we measured the levels of eight UPAHMs as biomarkers of exposure to avoid the uncertainty associated with single biomarkers.

The effects of PAHs generated during residential and professional cooking on platelet counts have been recorded. Our findings suggest that PAH exposure (in particular PAHs that metabolize to 9-OHFlu) affects the MPV, PDW, and P-LCR.

The participants in our study resided in the immediate coal plant area, where they were continuously exposed to higher levels of PAHs than citizens whose vulnerabilities originated from various other daily PAH exposure methods. The number of female participants was larger than that of male participants, which may have resulted from the inclusion and exclusion criteria. For more than 25 years, all participants had resided in a village located 1–2 km downwind, from a coal plant, but were not employed by the plant or related businesses. Therefore, a large number of men were excluded from the study because men were more likely to be employed by the coal plant or related businesses. Additionally, in the GLMM models, sex was added as a covariate and did not show a significant effect in these models. However, among children, males [31] had more DNA damage than females when exposed to waste incinerators. Regarding dietary consumption of contaminated vegetables, females had a higher exposure rate than males [32]. Exposure to second-hand smoke modified some of the associations, which may be because half of the participants were female and were more likely to be exposed to secondhand smoke. Cigarette smoke exposure (second-hand smoke) causes significantly elevated DNA damage among children [33, 34]. Although DNA damage has not been well-studied in adults, the effects of smoke on urinary levels of 1-OHNa, 2-OHNa, and 2-OHFlu have been reported by Cao et al. [2]. We plan to collect information from a larger cohort to validate the correlations with exposure to second-hand smoke.

Our findings were derived from longitudinal data analyzed using repeated measures. Generalized mixed models and generalized linear mixed models were fitted to adjust for participant covariates. These models have been used in previous studies. Yuan et al. [35] evaluated three models to assess the relationship between UPAHM levels and platelet indices. Similarly, we used three adjusted models with time as an additional covariate and assessed them using AIC and BIC. Armstrong and Gibbs also used AIC to evaluate the model fit, and Etemadi et al. constructed models that contained general information, genetic information, and environmental variables, with AICs ranging from 100.07 to 160.68. For our models, the AIC was

always negative, ranging from −340.6 to −1775.5. BIC is defined chiefly as signifying an accurate model, and is seldom applied in this context. However, we used BIC because our data were longitudinal [36] and included general linear mixed models [37]. This model selection process has been tested in studies of PAH bioaccessibility to plants [38].

To the best of our knowledge, no previous study had used human biomarkers of PAH exposure with a longitudinal dataset and repeated measures. However, some limitations should be noted. First, there were 418 participants enrolled in our study, of whom 222 participants were finally included and followed-up for 3 years. However, this may occur in various ways due to exposure to PAHs. The remaining 222 participants satisfied the limited PAH exposure criteria and completed the 3 years' follow up. We plan to prolong the follow-up period, to include more participants in further research. Second, the PAH exposure of each participant was difficult to estimate precisely. Our estimation could only provide a trend for each correlation, although we excluded the residents who had lived in the village for less than 25 years, and who had barbecue habits, and included smoking, second-hand smoke exposure, and alcohol use as covariates in Model 3. We would further improve our inclusion and exclusion criteria by increasing the number of participants. Finally, genetic information may also affect platelet indices. We will collect this information and add it as covariates in GLMMs in a future study.

## Conclusions

Long-term exposure to low concentrations of PAHs with a higher level of 2-OHPh, 1-OHPh, 1-OHP, has performed an increasing effect on the PLT. The 9-OHFlu has a rising impact on both PDW and MPV, by eliminating the influence of other ways of PAH exposures.

## Supporting information

**S1 Fig. Correlations among demographic information and urinary metabolite levels.** (TIF)

**S1 Table. The GMM model of PAH and blood routine indexes.** (DOCX)

**S2 Table. The GLMM model of PAH and PLT.** Notes, the confidence and 95% confidence interval result of the models have listed in this each cell of the table, by the order from Model 1 to Model 3. (DOCX)

**S3 Table. The GLMM model of 9-OHFlu and three platelet indices.** Notes, the confidence and 95% confidence interval result of the models have been listed in each cell of the table, by the order from Model 1 to Model 3. (DOCX)

**S4 Table. The variables collected from all 222 participants.** Note, we collected the variables from each of the 222 patients in these three aspects. The variables in the latter two aspects have been collected three times for repeated measures. While list their full names, their acronyms have been listed in the brackets for the convenience of further use. (DOCX)

**S5 Table. The variates of platelet indices and UPAHM among times.** Note, PLT is short for the count of platelet, while PDW for Platelet distribution width, MPV for mean platelet volume, PCT for platelet crit, P-LCR for large platelet ratio, 2-OHNa for 2-hydroxynaphthalene, 1-OHNa for 1-hydroxynaphthalene, 2-OHFlu for 2-hydroxyfluorene, 9-OHFlu for 9-hydroxyfluorene, 2-OHPh for 2-hydroxyphenanthrene, 1-OHPh for 1-hydroxyphenanthrene, 1-OHP

for 1-hydroxypyrene, 3-OHBaP for 3-hydroxybenzo[a]pyrene. Listing the geometric mean± geometric standard error for each following-up time of each index, the p values according to Mauchly's test of sphericity have been recorded in the fourth row of each group with "ST" for short. The last row of each group listed the results of ANOVA for repeated measurement (MANOVA) or traditional ANOVA analysis while their P values have been displayed in the brackets after F values. * stands for the statistical significance with $\alpha = 0.05$.
(DOCX)

**S6 Table. The minimal data set.**
(XLSX)

**S1 File. Generalized linear mixed model and generalized mixed model: The description of both modeling methods.**
(DOCX)

## Acknowledgments

We thank Liwen Bianji, Edanz Group China (www.liwenbianji.cn/ac), for editing the English text of a draft of this manuscript.

## Author Contributions

**Conceptualization:** Jing Cui, Caili Zhao, Jiefang Duan.

**Data curation:** Jing Cui, Ting Zhang, Chao Zhang, Zhenwei Xue, Xiaona Kong, Caili Zhao, Zimeng Li, Xiaoming Liu, Wenjie Peng.

**Formal analysis:** Chao Zhang, Zhenwei Xue.

**Funding acquisition:** Xiaoming Liu, Jiefang Duan, Wenjie Peng, Xiaolin Zhou.

**Investigation:** Zhenwei Xue, Xiaona Kong, Caili Zhao, Zimeng Li, Jiefang Duan, Wenjie Peng, Xiaolin Zhou, Hongmei Yu.

**Methodology:** Jing Cui, Xiaona Kong, Zimeng Li.

**Project administration:** Ting Zhang, Zhenwei Xue, Durong Chen, Xiaona Kong, Xiaoming Liu, Wenjie Peng.

**Resources:** Ting Zhang, Durong Chen, Xiaona Kong, Caili Zhao, Yufeng Guo, Xiaolin Zhou.

**Software:** Xiaona Kong, Caili Zhao, Jiefang Duan.

**Supervision:** Durong Chen, Yufeng Guo, Xiaoming Liu, Hongmei Yu.

**Validation:** Chao Zhang, Durong Chen, Yufeng Guo, Wenjie Peng, Hongmei Yu.

**Visualization:** Yufeng Guo, Hongmei Yu.

**Writing – original draft:** Jing Cui, Zimeng Li.

**Writing – review & editing:** Xiaolin Zhou, Hongmei Yu.

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
