## [Decision Letter · Decision Letter 0]

11 Aug 2022

PONE-D-22-17020Long-term exposure to low concentrations of polycyclic aromatic hydrocarbons and alterations in platelet indices: A longitudinal study in China

PLOS ONE

Dear Dr. Yu,

Thank you for submitting your manuscript to PLOS ONE. After careful consideration, we feel that it has merit but does not fully meet PLOS ONE’s publication criteria as it currently stands. Therefore, we invite you to submit a revised version of the manuscript that addresses the points raised during the review process.

We look forward to receiving your revised manuscript.

Kind regards,

Govarthanan Muthusamy

Academic Editor

PLOS ONE

Journal Requirements:

   "Yes, Pro. Xiaolin Zhou has supervised the writing process and the work has been approved by the Institutional Review Board of China Institute for Radiation Protection (CIRP). Ethics approval number: 21111011101EHSM(2019)SX-03."

Additional Editor Comments:

I have reviewed the manuscript entitled “Long-term exposure to low concentrations of polycyclic aromatic hydrocarbons and alterations in platelet indices: A longitudinal study in China”. The paper investigates the low-concentration exposure to PAHs and alterations in platelet indices in the Chinese population based on the longitudinal dataset using UPAHM as biomarkers of exposure. I wish to point out only some modifications to shape up the manuscript.

1. PAHs abbreviation must be explained at its first mention (in the abstract).

2. The abstract of the manuscript is required to be upgraded.

3. At an end of the introduction, authors should add in detail objective of the work.

4. Why authors selected more female participants? Any specific reason.

5. The authors should be uniformed the units and symbols according to journal format.

6. Improve the quality of the figures.

7. In discussion section, authors should cite recent references with more detail discussion.

8. Grammatical mistakes pervade the manuscript. I suggest the authors to carefully correct them.

9. References must be formatted according to the standard style of materials letters journal.

Reviewers' comments:

Reviewer's Responses to Questions

**Comments to the Author**

1. Is the manuscript technically sound, and do the data support the conclusions?

Reviewer #1: Yes

2. Has the statistical analysis been performed appropriately and rigorously? 

Reviewer #1: Yes

3. Have the authors made all data underlying the findings in their manuscript fully available?

Reviewer #1: Yes

4. Is the manuscript presented in an intelligible fashion and written in standard English?

Reviewer #1: Yes

5. Review Comments to the Author

Reviewer #1: I have reviewed the manuscript entitled “Long-term exposure to low concentrations of polycyclic aromatic hydrocarbons and alterations in platelet indices: A longitudinal study in China”. The paper investigates the low-concentration exposure to PAHs and alterations in platelet indices in the Chinese population based on the longitudinal dataset using UPAHM as biomarkers of exposure. I wish to point out only some modifications to shape up the manuscript.

1. PAHs abbreviation must be explained at its first mention (in the abstract).

2. The abstract of the manuscript is required to be upgraded.

3. At an end of the introduction, authors should add in detail objective of the work.

4. Why authors selected more female participants? Any specific reason.

5. The authors should be uniformed the units and symbols according to journal format.

6. Improve the quality of the figures.

7. In discussion section, authors should cite recent references with more detail discussion.

8. Grammatical mistakes pervade the manuscript. I suggest the authors to carefully correct them.

9. References must be formatted according to the standard style of materials letters journal.

6. PLOS authors have the option to publish the peer review history of their article (what does this mean?). If published, this will include your full peer review and any attached files.

Reviewer #1: No

---

## [Author Response · Author response to Decision Letter 0]

18 Sep 2022

Dear Dr. Govarthanan Muthusamy;

We thank the Editors and Reviewers for their accurate and insightful comments, and for the careful attention that they have paid to our manuscript. We have carefully revised the manuscript in response to all the comments. Our point-by-point responses to the Editors and Reviewers’ comments are listed below. We hope that this new draft addresses all of your concerns, and that our manuscript is now suitable for publication in your journal.

Sincerely,

Hongmei Yu

Reviewer #1:

Comment 1: PAHs abbreviation must be explained at its first mention (in the abstract).

Response: We thank the Reviewer for pointing this out. We added the PAHs abbreviation. Specific revisions are as follows.

“Abstract

Long-term exposure to low polycyclic aromatic hydrocarbon (PAH) concentration may ave detrimental effects, including changing platelet indices. ”

Comment 2: The abstract of the manuscript is required to be upgraded.

Response: We thank the Reviewer for this constructive comment. We have carefully revised the abstract. Specific revisions are as follows.

“Abstract(page 2)

Long-term exposure to low polycyclic aromatic hydrocarbon (PAH) concentration may ave detrimental effects, including changing platelet indices. Effects of chronic exposure to low PAH concentrations have been evaluated in cross-sectional, but not in longitudinal studies, to date. We aimed to assess the effects of long-term exposure to the low-concentration PAHs on alterations in platelet indices in the Chinese population. During 2014 - 2017, we enrolled 222 participants who had lived in a village in northern China, 1-2 km downwind from a coal plant, for more than 25 years, but who were not employed by the plant or related businesses. During three follow-ups, annually in June, demographic information and urine and blood samples were collected. Eight PAHs were tested: namely 2-hydroxynaphthalene, 1-hydroxynaphthalene, 2-hydroxyfluorene, 9-hydroxyfluorene (9-OHFlu), 2-hydroxyphenanthrene (2-OHPh), 1-hydroxyphenanthrene (1-OHPh), 1-hydroxypyrene (1-OHP), and 3-hydroxybenzo [a] pyrene. Five platelet indices were measured: platelet count (PLT), platelet distribution width (PDW), mean platelet volume (MPV), platelet crit, and the platelet-large cell ratio. Generalized mixed and generalized linear mixed models were used to estimate correlations between eight urinary PAH metabolites and platelet indices. Model 1 assessed whether these correlations varied over time. Models 2 and 3 adjusted for additional personal information and personal habits. We found the following significant correlations: 2-OHPh (Model1 β_1 = 18.06，Model2 β_2 = 18.54, Model3 β_3 = 18.54), 1-OHPh (β_1 = 16.43, β_2 = 17.42, β_3 = 17.42), 1-OHP (β_1 = 13.93, β_2 = 14.03, β_3 = 14.03) with PLT, as well as 9-OHFlu with PDW and MPV (odds ratio or Model3 〖OR〗_PDW [95%CI] = 1.64[1.3-2.06], 〖OR〗_MPV [95%CI] = 1.33[1.19-1.48]). Long-term exposure to low concentrations of PAHs, indicated by2-OHPh, 1-OHPh, 1-OHP, and 9-OHFlu, as urinary biomarkers, affects PLT, PDW, and MPV. 9-OHFlu increased both PDW and MPV after elimination of the effects of other PAH exposure modes.

Comment 3: At an end of the introduction, authors should add in detail objective of the work.

Response: We thank the Reviewer for this insightful comment. We have carefully revised the introduction. Specific revisions are as follows.

“Introduction (page 5, paragraph 1)

Thus, we assessed the effects of long-term exposure to low-concentration PAHs and alterations in platelet indices in the Chinese population. Longitudinal datasets were collected to study the effects of long-term exposure to low concentrations of PAHs on platelet indices among adults who were not occupationally exposed. We then estimated the correlations among eight representative urinary PAH metabolites (UPAHMs) and five platelet indices, under the effect of covariates, aiming to assess low-concentration exposure to PAHs and alterations in platelet indices in the Chinese population based on a longitudinal dataset using UPAHM as biomarkers of exposure.”

Comment 4: Why authors selected more female participants? Any specific reason.

Response: We thank the Reviewer for pointing this out. In the Methods section, as we pointed out in the original manuscript, all participants had resided for more than 25 years in a village located downwind and 1-2 km from a coal plant but were not employed by the plant or related businesses. Therefore, large numbers of men were excluded from the study because men are more likely to be employed by the plant or related businesses. Table 1 shows gender did not differ by year of recruitment into the study. We added this into the Discussion section of our manuscript.

“Discussion (page 16, paragraph 3)

The number of female participants was larger than that of male participants, which may have resulted from the inclusion and exclusion criteria. For more than 25 years, all participants had resided in a village located 1–2 km downwind, from a coal plant, but were not employed by the plant or related businesses. Therefore, a large number of men were excluded from the study because men were more likely to be employed by the coal plant or related businesses.”

Comment 5: The authors should be uniformed the units and symbols according to journal format.

Response: We thank the Reviewer for pointing this out. We have carefully added the units in Table 2 and 3, while change the symbols of paragraph after Table 3.

Table 2. The fixed effect among platelet indices and UPAHM based on GMMs

UPAHM PLT/F1(%) PDW(fL) MPV(fL) PCT(%) P-LCR(%)

2-OHNa(ng/g·cr-1) 0.34 (0.5627) 3.03 (0.0826) 3.07 (0.0806) 2.36 (0.1253) 0.21 (0.6497)

1-OHNa(ng/g·cr-1) 5.22 (0.0228)* 0.23 (0.6332) 0.13 (0.7201) 0.44 (0.5060) 0.10 (0.7527)

9-OHFlu(ng/g·cr-1) 3.75 (0.0534) 6.01 (0.0146)* 6.26 (0.0127)* 0.01 (0.9270) 4.37 (0.0372)*

2-OHFlu(ng/g·cr-1) 5.45 (0.0200)* 0.21 (0.6469) 0.42 (0.5165) 0.53 (0.4652) 1.20 (0.2744)

2-OHPh(ng/g·cr-1) 4.71 (0.0306)* 0.16 (0.6922) 0.22 (0.6399) 1.74 (0.1884) 2.10 (0.1480)

1-OHPh(ng/g·cr-1) 5.45 (0.0200)* 0.31 (0.5761) 0.41 (0.5227) 0.77 (0.3806) 0.62 (0.4314)

1-OHP(ng/g·cr-1) 6.17 (0.0134)* 0.30 (0.5862) 0.16 (0.6854) 0.67 (0.4133) 2.65 (0.1040)

3-OHBaP(ng/g·cr-1) 0.01 (0.9050) 1.57 (0.2112) 0.77 (0.3811) 2.03 (0.1549) 0.61 (0.4339)

Table 3. Dose-reposed effect of PAH exposure on PLT

Level of UPAHM Model 1 Model 2 Model 3

1-OHNa(ng/g·cr-1) 

Con(β[P value]) 6.65[0.052] 6.54[0.057] 6.54[0.057]

1 0.95(0.916-0.986)* 0.951(0.916-0.987)* 0.95(0.916-0.986)*

2 0.945(0.9-0.993)* 0.946(0.9-0.993)* 0.943(0.897-0.991)*

3 0.962(0.924-1.001) 0.962(0.924-1.002) 0.961(0.923-1.001)

4 1 1 1

AIC -286.9 -268.9 -251

BIC -266.5 -248.5 -230.6

2-OHFlu(ng/g·cr-1) 

Con(β[P value]) 22.37[<0.001*] 22.8[<0.001*] 22.8[<0.001*]

1 0.966(0.928-1.005) 0.966(0.928-1.006) 0.967(0.928-1.007)

2 0.961(0.923-1) 0.961(0.922-1) 0.961(0.923-1.001)

3 1.008(0.969-1.049) 1.007(0.967-1.049) 1.007(0.967-1.049)

4 1 1 1

AIC -286.2 -268.2 -249.6

BIC -265.8 -247.8 -229.2

2-OHPh(ng/g·cr-1) 

Con(β[P value]) 18.06[<0.001*] 18.54[<0.001*] 18.54[<0.001*]

1 0.941(0.904-0.979)* 0.941(0.904-0.98)* 0.941(0.904-0.98)*

2 0.939(0.904-0.976)* 0.939(0.904-0.977)* 0.94(0.904-0.977)*

3 0.951(0.912-0.991)* 0.95(0.911-0.99)* 0.949(0.91-0.99)*

4 1 1 1

AIC -290.1 -272.2 -253.8

BIC -269.7 -251.8 -233.4

1-OHPh(ng/g·cr-1) 

Con(β[P value]) 16.43[<0.001*] 17.42[<0.001*] 17.42[<0.001*]

1 0.958(0.92-0.997)* 0.958(0.92-0.997)* 0.957(0.919-0.996)*

2 0.958(0.924-0.993)* 0.958(0.924-0.994)* 0.958(0.924-0.994)*

3 0.992(0.944-1.042) 0.991(0.943-1.041) 0.99(0.942-1.04)

4 1 1 1

AIC -285.4 -267.5 -249.2

BIC -265 -247.1 -228.8

1-OHP(ng/g·cr-1) 

Con(β[P value]) 13.93[0.001*] 14.03[0.001*] 14.03[0.001*]

1 0.948(0.902-0.996)* 0.949(0.903-0.997)* 0.949(0.903-0.997)*

2 0.99(0.942-1.04) 0.992(0.944-1.043) 0.992(0.944-1.043)

3 0.988(0.942-1.035) 0.989(0.944-1.037) 0.99(0.944-1.038)

4 1 1 1

AIC -285.4 -267.8 -249.4

BIC -265 -247.3 -228.9

“Model 1: β1[P value]= 0.48 [< 0.001], Model 2 β2[P value]= 0.49 [< 0.001], Model 3 β3[P value]= 0.49 [< 0.001]) correlated with PDW and with MPV (β1[P value]= 0.28 [< 0.001], β2[P value]= 0.28 [< 0.001], β3[P value]= 0.28 [< 0.001]), and with P-LCR (β1[P value]= 0.58 [0.458], β2[P value]= 0.57 [0.463], β3[P value]= 0.57 [0.463]) in all three models, all of which had an increasing effect on platelet indices.”

“For MPV, participants without exposure to second-hand smoke exhibited a dose–response trend at the second and third levels, in which the second exposure level (ORwithout =0.958; 95%CI [0.924–0.994]) and the second exposure level (ORwithout= 1.049; 95%CI[1.013–1.086]) differed from the highest exposure level.”

Comment 6: Improve the quality of the figures.

Response: We thank the Reviewer for this constructive comment. We re-uploaded the clear figures. And changed them into PDF documents.

Comment 7: In discussion section, authors should cite recent references with more detail discussion.

Response: We thank the Reviewer for pointing this out. We have carefully revised the discussion.

“

28. Zhang, H., L. Yuan, J. Xue and H. Wu, Polycyclic aromatic hydrocarbons in surface water and sediment from Shanghai port, China: spatial distribution, source apportionment, and potential risk assessment. Environ Sci Pollut Res Int, (2022). doi:10.1007/s11356-022-22706-5.

29. Sopian, N.A., J. Jalaludin, S. Abu Bakar, T.R. Hamedon and M.T. Latif, Exposure to Particulate PAHs on Potential Genotoxicity and Cancer Risk among School Children Living Near the Petrochemical Industry. Int J Environ Res Public Health, (2021). 18. doi:10.3390/ijerph18052575.

30. Oliveira, M., K. Slezakova, C. Delerue-Matos, M.C. Pereira and S. Morais, Children environmental exposure to particulate matter and polycyclic aromatic hydrocarbons and biomonitoring in school environments: A review on indoor and outdoor exposure levels, major sources and health impacts. Environ Int, (2019). 124: p. 180-204. doi:10.1016/j.envint.2018.12.052.

31. Xu, P., Z. Chen, Y. Chen, L. Feng, L. Wu, D. Xu, et al., Body burdens of heavy metals associated with epigenetic damage in children living in the vicinity of a municipal waste incinerator. Chemosphere, (2019). 229: p. 160-168. doi:10.1016/j.chemosphere.2019.05.016.

32. Ailijiang, N., X. Cui, A. Mamat, Y. Mamitimin, N. Zhong, W. Cheng, et al., Levels, source apportionment, and risk assessment of polycyclic aromatic hydrocarbons in vegetable bases of northwest China. Environ Geochem Health, (2022). doi:10.1007/s10653-022-01369-8.

33. Zalata, A., S. Yahia, A. El-Bakary and H.M. Elsheikha, Increased DNA damage in children caused by passive smoking as assessed by comet assay and oxidative stress. Mutat Res, (2007). 629: p. 140-7. doi:10.1016/j.mrgentox.2007.02.001.

34. Beyoglu, D., T. Ozkozaci, N. Akici, G.Z. Omurtag, A. Akici, O. Ceran, et al., Assessment of DNA damage in children exposed to indoor tobacco smoke. Int J Hyg Environ Health, (2010). 213: p. 40-3. doi:10.1016/j.ijheh.2009.10.001.

35. Chunjie, Yuan, Jian, Hou, Yun, Zhou, et al., Dose-response relationships between polycyclic aromatic hydrocarbons exposure and platelet indices. Environmental Pollution, (2018).

36. Jones, R.H., Bayesian information criterion for longitudinal and clustered data. Stats in Medicine, (2011). 30.

37. Sanquetta, C.R., A.P.D. Corte, A. Behling, L.R.D.O. Piva and M.N.I. Sanquetta, Selection criteria for linear regression models to estimate individual tree biomasses in the Atlantic Rain Forest, Brazil. Carbon Balance Management, (2018). 13.”

Comment 8: Grammatical mistakes pervade the manuscript. I suggest the authors to carefully correct them.

Response: We thank the Reviewer for pointing this out. This new manuscript has been edited extensively by professional editing group.

Comment 9: References must be formatted according to the standard style of materials letters journal.

Response: We thank the Reviewer for pointing this out. We have edited the style of the reference.

Journal Requirements:

Comment 1: Please ensure that your manuscript meets PLOS ONE's style requirements, including those for file naming.

Response: We thank the Editors for pointing this out. We have edited the names of Table and Figure(eg.Fig 1; S1 Table).

Comment 2: Please provide additional details regarding participant consent.

Response: We thank the Editors for pointing this out. We have added the ethics approval and the informed consent form as S2 and S3 Figs.

Comment 3: We note that the grant information you provided in the ‘Funding Information’ and ‘Financial Disclosure’ sections do not match. 

Response: We thank the Editors for pointing this out. And we make the consistency of funding number. The work has been approved by the Institutional Review Board of China Institute for Radiation Protection (CIRP). The funding number is 21111011101EHSM(2019)SX-03. 

Comment 4: Please state what role the funders took in the study. 

Response: We thank the Editors for pointing this out. The funders had no role in study design, data collection and analysis, decision to publish, or preparation of the manuscript. And add this into the cover letter.

Comment 5: PLOS journals require that the minimal data set be made fully available.

Response: We thank the Editors for pointing this out. We have added the minimal data set as S6 Table.

Comment 6: PLOS requires an ORCID ID for the corresponding author in Editorial Manager on papers submitted

Response: We thank the Editors for pointing this out. We have log in and connected the ORCID ID of corresponding author.

Comment 7: Please include captions for your Supporting Information files at the end of your manuscript, and update any in-text citations to match accordingly.

Response: We thank the Editors for pointing this out. We have carefully revised the part.

“Supporting information 

S1 Fig. Correlations among demographic information and urinary metabolite levels 

S2 Fig. Ethics approval 

S3 Fig. Informed consent form

S4 Fig. language editing certification

S1 Table. The GMM Model of PAH and blood routine indexes. 

S2 Table. The GLMM Model of PAH and PLT. Notes, the confidence and 95% confidence interval result of the models have listed in this each cell of the table, by the order from Model 1 to Model 3. 

S3 Table. The GLMM Model of 9-OHFlu and three platelet indices. Notes, the confidence and 95% confidence interval result of the models have been listed in each cell of the table, by the order from Model 1 to Model 3. 

S4 Table. The variables collected from all 222 participants. Note, we collected the variables from each of the 222 patients in these three aspects. The variables in the latter two aspects have been collected three times for repeated measures. While list their full names, their acronyms have been listed in the brackets for the convenience of further use. 

S5 Table. The variates of platelet indices and UPAHM among times. Note, PLT is short for the count of platelet, while PDW for Platelet distribution width, MPV for mean platelet volume, PCT for platelet crit, P-LCR for large platelet ratio, 2-OHNa for 2-hydroxynaphthalene, 1-OHNa for 1-hydroxynaphthalene, 2-OHFlu for 2-hydroxyfluorene, 9-OHFlu for 9-hydroxyfluorene, 2-OHPh for 2-hydroxyphenanthrene, 1-OHPh for 1-hydroxyphenanthrene, 1-OHP for 1-hydroxypyrene, 3-OHBaP for 3-hydroxybenzo[a]pyrene. Listing the geometric mean± geometric standard error for each following-up time of each index, the p values according to Mauchly's test of sphericity have been recorded in the fourth row of each group with "ST" for short. The last row of each group listed the results of ANOVA for repeated measurement (MANOVA) or traditional ANOVA analysis while their P values have been displayed in the brackets after F values. * stands for the statistical significance with α=0.05.

S6 Table. The minimal data set

S1 file. Generalized Linear Mixed Model and Generalized Mixed Model: The description of both modeling methods.”

Comment 8: Please review your reference list to ensure that it is complete and correct.

Response: We thank the Editors for pointing this out. We have carefully revised the reference part.

---

## [Decision Letter · Decision Letter 1]

18 Oct 2022

Long-term exposure to low concentrations of polycyclic aromatic hydrocarbons and alterations in platelet indices: A longitudinal study in China

PONE-D-22-17020R1

Dear Dr. Yu,

We’re pleased to inform you that your manuscript has been judged scientifically suitable for publication and will be formally accepted for publication once it meets all outstanding technical requirements.

Kind regards,

Govarthanan Muthusamy

Academic Editor

PLOS ONE

Additional Editor Comments (optional):

Reviewers' comments:

Reviewer's Responses to Questions

**Comments to the Author**

1. If the authors have adequately addressed your comments raised in a previous round of review and you feel that this manuscript is now acceptable for publication, you may indicate that here to bypass the “Comments to the Author” section, enter your conflict of interest statement in the “Confidential to Editor” section, and submit your "Accept" recommendation.

Reviewer #1: All comments have been addressed

2. Is the manuscript technically sound, and do the data support the conclusions?

Reviewer #1: Yes

3. Has the statistical analysis been performed appropriately and rigorously? 

Reviewer #1: Yes

4. Have the authors made all data underlying the findings in their manuscript fully available?

Reviewer #1: Yes

5. Is the manuscript presented in an intelligible fashion and written in standard English?

Reviewer #1: Yes

6. Review Comments to the Author

Reviewer #1: The manuscript has been improved after revision, according to the reviewer's comments. I recommend the paper to be published.

7. PLOS authors have the option to publish the peer review history of their article (what does this mean?). If published, this will include your full peer review and any attached files.

Reviewer #1: No

---

## [Editor Report · Acceptance letter]

24 Oct 2022

PONE-D-22-17020R1 

Long-term exposure to low concentrations of polycyclic aromatic hydrocarbons and alterations in platelet indices: A longitudinal study in China 

Dear Dr. Yu:

I'm pleased to inform you that your manuscript has been deemed suitable for publication in PLOS ONE. Congratulations! Your manuscript is now with our production department. 

Kind regards, 

on behalf of

Dr. Govarthanan Muthusamy 

Academic Editor

PLOS ONE